# Microvascular Changes after Scleral Buckling for Rhegmatogenous Retinal Detachment: An Optical Coherence Tomography Angiography Study

**DOI:** 10.3390/diagnostics12123015

**Published:** 2022-12-01

**Authors:** Matteo Fallico, Antonio Longo, Teresio Avitabile, Pietro Alosi, Marco Grillo, Niccolò Castellino, Vincenza Bonfiglio, Michele Reibaldi, Francesco Pignatelli, Andrea Russo

**Affiliations:** 1Department of Ophthalmology, University of Catania, 95123 Catania, Italy; 2Department of Experimental Biomedicine and Clinical Neuroscience, Ophthalmology Section, University of Palermo, 90127 Palermo, Italy; 3Department of Surgical Sciences, Eye Clinic Section, University of Turin, 10122 Turin, Italy; 4Eye Clinic, Hospital “SS Annunziata”, ASL Taranto, 74010 Taranto, Italy

**Keywords:** scleral buckling, retinal detachment, optical coherence tomography angiography, vitreoretinal surgery, vessel density, superficial capillary plexus, deep capillary plexus

## Abstract

This retrospective study aimed to investigate macular microvascular alterations after successful scleral buckling (SB) for rhegmatogenous retinal detachment (RRD). Nineteen eyes with macula-on RRD and 18 eyes with macula-off RRD were included. In all cases, an encircling band was placed. Optical coherence tomography angiography (OCTA) was performed at baseline and postoperatively. Changes in the foveal avascular zone (FAZ) area and vessel density (VD) of the superficial capillary plexus (SCP) and deep capillary plexus (DCP) were the primary outcomes. Correlations between OCTA findings and clinical variables were considered secondary outcomes. In both the macula-on and macula-off groups, the FAZ area was comparable with controls. In the macula-on group, VD in the whole SCP was lower compared with controls at both baseline (*p* < 0.001) and 6 months (*p* = 0.03), but showed a significant increase after surgery (*p* = 0.004). In the macula-off group, postoperative VD in both whole SCP and whole DCP was lower compared with controls (*p* < 0.001). In the macula-on group, there was an inverse correlation between axial length increase and SCP VD change (r = −0.508; *p* = 0.03). These findings demonstrated microvascular alterations after SB for RRD. However, VD impairment seems to be mitigated after surgery. A greater increase in postoperative axial length was associated with a poorer VD outcome.

## 1. Introduction

Scleral buckling (SB) is a valuable treatment option for rhegmatogenous retinal detachment (RRD), being preferred over vitrectomy in cases of phakic eyes with uncomplicated or medium complexity detachments [1,2]. Most commonly, surgery involves the use of an encircling band, placed posteriorly to the vitreous base, aimed at relieving vitreoretinal traction and protecting the vitreous base from degenerative processes [3,4]. Circumferential buckles, particularly encircling bands, cause significant changes to eyeball anatomy secondary to their pressure exerted on the sclera. The ultimate consequence is an increase in axial length [5]. These anatomical changes, in turn, might cause microvascular alterations [6,7].

Optical coherence tomography angiography (OCTA) is a novel, noninvasive imaging method that provides qualitative and quantitative information on macular microcirculation [8]. OCTA has recently been used to evaluate macular microvascular changes that occur after successful RRD surgery [9,10,11,12,13,14,15,16,17]. In particular, many authors have investigated microvascular alterations following vitrectomy surgery, while less attention has been paid to the effect of scleral buckling on macular microvascular conditions.

The purpose of this study was to evaluate macular vessel density (VD) on OCTA following scleral buckling for RRD and to identify which clinical variables might have an influence on macular microvascular status. Additionally, the relationships between microvascular parameters and functional and anatomical outcomes were explored.

## 2. Materials and Methods

This retrospective study was conducted at the Eye Clinic of the University of Catania, Italy. The study protocol was in agreement with the tenets of the Declaration of Helsinki and was approved by the Institutional Review Board (number 69/2021/PO). Informed consent was obtained from all subjects. Medical charts of all consecutive patients who underwent scleral buckling surgery for primary RRD between September 2017 and December 2020 were reviewed.

For inclusion, the following criteria had to be satisfied: (a) scleral buckling procedure as primary surgery for RRD in a phakic eye with healthy fellow eye; (b) follow-up of 6 months or longer; (c) anatomically reattached retina following single surgery with no evidence of intraoperative and postoperative complications; (d) spectral domain-OCT (sd-OCT) and OCTA imaging at baseline and throughout the follow-up, namely at 1 month (±7 days), 3-month (±14 days) and 6-month (±14 days) post-operatively; (e) absence of abnormalities on sd-OCT imaging, such as intraretinal cysts, epiretinal membrane, intraretinal or subretinal spaces, external limiting membrane disruption, ellipsoid zone disruption, and retinal pigment epithelium alterations. Subjects with a history of ophthalmological diseases that could affect visual acuity and vascular density, such as trauma, amblyopia, glaucoma, macular degeneration, macular hole, retinopathy of any type, uveitis and pathological myopia (axial length >26 mm and/or a spherical equivalent ≥−6 diopters) were excluded. Patients with a history of any prior intraocular surgery, except uncomplicated cataracts, in either eye were also excluded. These exclusion criteria were applied to the fellow eye as well and patients with an anisometropia >2.0 Diopters were excluded since the fellow eye was used as the control eye.

All SB surgeries were performed by the same vitreoretinal surgeon under retrobulbar anesthesia. Following a 360° conjunctival peritomy and rectus muscle isolation, all retinal tears were localized. A 2.4 mm encircling silicone band (Mira^®^, Mira Inc., Uxbridge, MA, USA) was placed under each rectus muscle. Segmental buckles were inserted underneath the silicone band to ensure retinal tear closure. Evacuative puncture and trans-scleral cryopexy were performed in all cases.

All patients received a complete eye examination at baseline, which means before SB surgery, and at each follow-up visit, namely at 1, 3 and 6 months postoperatively. Eye examinations included best-corrected visual acuity (BCVA) measurement, slit-lamp examination, dilated fundus examination, and Goldmann tonometry. The BCVA was evaluated using Early Treatment Diabetic Retinopathy charts and converted into the logarithm of the minimum angle of resolution units (logMAR).

Axial length was measured preoperatively and 6-month post-operatively by using an IOL Master device (Carl Zeiss Meditec, Dublin, CA, USA) in eyes with macula-on RRD, while an immersion ultrasound biometry (Quantel Compact Touch, Quantel Medical, TX, USA) was performed in eyes with macula-off RRD.

Spectral domain OCT and OCTA were performed using the XR Avanti AngioVue System (Optovue Inc., Fremont, CA, USA). A 6 × 6 mm high-definition (400 × 400) Angio scan pattern, centered on the foveola, was carried out in both eyes at baseline and 1, 3 and 6 months postoperatively. This imaging is based on the split-spectrum amplitude decorrelation angiography (SSADA) method. AngioAnalytic software (Optovue, Inc., Fremont, CA, USA) automatically calculated the VD in the superficial capillary plexus (SCP) and deep capillary plexus (DCP) of the scanned area, providing these data for the whole image as well as for the foveal, parafoveal (3 mm) and perifoveal (6 mm) ETDRS sub-fields. VD refers to the proportion of the vessel area in the region of interest. The foveal avascular zone (FAZ) area was automatically analyzed. Central macular thickness (CMT) was recorded using the same OCT system. The inbuilt three-dimensional (3D) projection artifact removal (PAR) algorithm improved the depth resolution of vascular layers and distinguished vascular plexus-specific features. The inbuilt software automatically evaluated scan quality and signal strength index (SSI). Two investigators (M.F. and A.L.) independently reviewed OCT and OCTA scans of eligible patients: patients with a scan quality <6 and/or an SSI < 60 and/or residual motion artifacts were excluded. In cases of disagreement in the imaging review process, a third investigator (A.R.) was consulted. Scan segmentation was also checked, and when observed, errors were corrected using the inbuilt editing and propagation tools.

For each included patient, demographic characteristics and clinical and OCTA data were collected. The included patients were divided into two subgroups according to the preoperative status of the macula: the macula-on group and the macula-off group. The primary outcome was to assess changes in OCTA VD and FAZ area following SB surgery in the two groups. In the macula-on group, the change between preoperative and postoperative data of the RRD eye was analyzed, while in the macula-off group, given the lack of reliable baseline scans in the RRD eye, only the postoperative change from month 1 to month 6 was analyzed. In both groups, these data were also compared with baseline data of the fellow eye, which was used as a control. BCVA, CMT and axial length changes were considered secondary outcomes in both groups. In the macula-on group, possible correlations between OCTA parameters (FAZ and VD) and changes in BCVA and axial length were also investigated as secondary outcomes.

### Statistical Analysis

In the two groups, the values of each parameter detected at different time points were compared by ANOVA; if significant, multiple comparisons were made using Tukey’s HSD test. A *t*-test was used to compare continuous variables between the RRD eyes and the control eyes. In the macula-on group, correlations were studied by applying the Pearson regression test. A *p* value < 0.05 was considered significant. SPSS Statistics software version 21 (IBM Corp. Armonk, NY, USA) was used.

## 3. Results

A total of 37 eyes were included, of which 19 were in the macula-on group and 18 in the macula-off group. Baseline demographics and ocular characteristics are shown in Table 1.

### 3.1. Macula-On RRD Group

The clinical and OCTA parameters of the macula-on RRD group are summarized in Table 2. Baseline mean BCVA was 0.10 ± 0.03 logMar and remained unchanged throughout the follow-up. Mean axial length increased from a baseline value of 25.2 ± 0.5 mm to 26.1 ± 0.8 mm at 6 months (*p* < 0.001). The mean baseline FAZ area was 0.227 ± 0.02 mm2 and remained unchanged following SB surgery. The mean baseline CMT was 244 ± 13 microns, with no significant change following SB surgery. Both the mean FAZ area and CMT in the RRD eyes were comparable with controls. Baseline VD in the whole SCP was lower in RRD eyes compared with controls (*p* < 0.001). A significant increase of VD in the whole SCP was demonstrated in RRD eyes following SB surgery (ANOVA, *p* = 0.004); however, 6-month VD in the whole SCP was lower compared with controls (*p* = 0.03). Similarly, SCP VD increased during follow-up in the parafoveal and perifoveal subfields (parafoveal SCP, *p* = 0.005; perifoveal SCP, *p* = 0.001), while it remained unchanged in the foveal subfield (*p* = 0.07). VD in the foveal SCP was comparable between RRD eyes and controls at both baseline (*p* = 0.83) and 6 months postoperatively (*p* = 0.74). In parafoveal and perifoveal subfileds of the SCP, VD was lower in RRD eyes compared with controls at both baseline (parafoveal SCP, *p*< 0.001; perifoveal SCP, *p*< 0.001) and 6 months postoperatively (parafoveal SCP, *p* = 0.02; perifoveal SCP, *p* = 0.04). VD in the DCP did not show any significant change following SB surgery and was comparable with the controls. Figure 1 shows baseline and postoperative OCTA imaging in a case of macula on RRD.

Axial length change was shown to correlate with mean change in VD of the whole SCP (r = −0.508, *p* = 0.03), perifoveal SCP (r = −0.546, *p* = 0.02) and parafoveal SCP (r = −0.684, *p* = 0.001). In all cases, an increase in axial length was associated with poorer VD outcomes (Figure 2). No other significant correlations were found.

### 3.2. Macula-Off RRD Group

The clinical and OCTA parameters of the macula-off RRD group are summarized in Table 3. Baseline mean BCVA improved following SB surgery, from 1.04 ± 0.09 logMar at baseline to 0.25 ± 0.08 logMar at 6 months (*p* < 0.001). The mean axial length increased from 24.9 ± 0.5 mm to 26.2 ± 0.7 mm at 6 months (*p* = 0.001). Baseline OCTA parameters were not available. Following SB surgery, from month 1 to month 6, no significant change in either the mean CMT or mean FAZ area was found. Both 6-month CMT and FAZ area were comparable with controls (FAZ, *p* = 0.26; CMT, *p* = 0.18).

Following SB surgery, from month 1 to month 6, no significant change in the VD of the SCP was shown. At 6 months, the mean VD of the whole SCP was lower compared with controls (*p* < 0.001). At 6 months, a lower-than-control VD was found in the parafoveal SCP and perifoveal SCP as well (parafoveal SCP, *p* < 0.001; perifoveal SCP, *p* < 0.001).

During the follow-up, from month 1 to month 6, VD significantly increased in the whole DCP (*p* < 0.001). Such a significant increase was found in the parafoveal DCP as well (*p* < 0.001). No VD change was seen in the other subfields of the DCP. At 6 months, mean VD in DCP was lower compared with controls (whole DCP, *p* < 0.001; foveal DCP, *p* = 0.004, parafoveal DCP, *p* < 0.001; perifoveal DCP, *p* < 0.001). Figure 3 shows baseline and postoperative OCTA imaging in a case of macula-off RRD.

## 4. Discussion

This study sought to investigate macular microvascular changes on OCTA imaging after SB for RRD, showing a postoperative reduction in the VD of the SCP in both macula-on and macula-off RRDs. The postoperative VD of DCP seems to be impaired in eyes affected by macula-off RRD, while it seems to be unchanged in macula-on RRDs.

Microvascular alterations in retinal and choroidal circulation occurring in RRD eyes have long been studied [6,9,18]. Before the advent of OCTA, other tools were used [6,18]. Scanning laser Doppler flowmetry showed a decrease in macular flow in the RRD eyes [18]. Such vascular changes are supposed to be related to reversible vasoconstriction, which could be secondary to tissue hypoxia [9,19]. In macula-off detachments, the biological composition of subretinal fluid has been assumed to affect oxygen diffusion due to an increase in inflammatory mediators [9,12,20]. Consequently, microvascular changes and photoreceptor alterations are likely to occur in the macula and might not be fully restored even after anatomical re-attachment [10,14,21]. These alterations, in turn, may determine functional defects, such as suboptimal visual recovery, color vision impairment, and persistent metamorphopsia [9,21,22].

The introduction of OCTA imaging represents a significant breakthrough in the assessment of retinal microcirculation. Several studies have explored macular microvascular alterations in OCTA imaging following RRD repair surgery [10,11,12,13,14,15,16,17,23]. The most available evidence is for vitrectomy cases [12,14,15,16,17,23]. A few reports have included both SB and vitrectomy cases. Barca et al. found augmentation in the FAZ area in the SB group compared with the vitrectomy group [10]. Tsen et al. found no difference in the FAZ area and VD between eyes with and without buckles [11]. Nam et al. found a greater VD reduction in vitrectomy cases compared with SB cases [13]. However, to the best of our knowledge, no study has specifically focused on microvascular alterations in OCTA following SB for RRD.

According to the majority of studies on vitrectomy cases, the VD of macular capillary plexi, both deep and superficial, is reduced after RRD surgery [10,11,13,14,16,17,23]. Hong et al. analyzed the macular VD of 31 eyes 6 months after vitrectomy for RRD, including both macula-on and macula-off cases [15]. Their study showed no difference in the mean VD of either DCP or SCP between RRD eyes and controls. These findings seem to be in disagreement with those reported by most studies but need to be looked at in more detail. A possible explanation for these controversial results is the fact that VD seems to increase in a long-term follow-up. Some authors have described a significantly reduced VD in the early post-operative period, which progressively increased throughout long-term follow-up [10,13]. Indeed, VD seems to be comparable between RRD eyes and controls between 6 and 12 months post-operatively [10,13]. Such a tendency could explain why Hong et al. [15] were unable to prove microvascular alterations at 6 months after RRD surgery.

Our findings showed that VD featured a similar increasing trend over a 6-month follow-up after SB surgery for RRD. In eyes with macula-on RRD, baseline VD was reduced in the SCP compared with controls; following the SB procedure, SCP VD increased over a 6-month period. In eyes with macula-off RRD, baseline OCTA was not available; in this group, VD in the DCP significantly improved from month 1 to month 6 postoperatively.

Barca et al. hypothesized that the impairment of VD of the SCP in eyes with macula-on RRD might be related to the fact that the SCP is the first vascular layer affected in the case of a rapid increase of vascular resistance secondary to a recent retinal detachment [10]. SCP is characterized by a greater density of smooth muscles and arterioles compared with the deep plexus. Consequently, vasoconstriction driven by tissue hypoxia is supposed to be stronger in the superficial plexus [10]. The improvement of SCP VD following successful RRD repair seems to confirm this hypothesis. Our findings in the macula-on cohort corroborated this. Furthermore, our correlation analysis between axial length change and VD provides new insight into the pathogenetic mechanism leading to microvascular alterations. In the macula-on cohort, the mean change in axial length was negatively correlated with the mean change in SCP VD. This means that the greater the postoperative increase in axial length, the deeper the reduction in VD. A greater increase in axial length is secondary to a tighter encircling band. The SB procedure included an evacuative puncture with external fluid drainage in all cases. The encircling band is likely to be tighter in cases with a greater amount of subretinal fluid because post-drainage hypotony is usually dealt with by tightening the band. On this basis, our assumption is that a greater increase in axial length could be associated with a larger amount of subretinal fluid. Consequently, we might speculate that the larger the amount of subretinal fluid, the greater the impairment of SCP VD. This corroborates the hypothesis of vasoconstriction in the macular SCP secondary to an increase in vascular resistance due to peripheral hypoxia in the detached retina. However, we cannot rule out that this inverse correlation between axial length increase and VD change could be a consequence of mechanical damage to the retinal microvascular network caused by the surgery itself. D’Aloisio et al. reported a reduction in iris perfusion on the anterior segment OCTA SB procedure [7]. The authors speculated that these alterations could be related to mechanical stress on vessels due to surgical maneuvers and encircling band application [7].

We found a reduction in VD in both superficial and deep capillary plexi in the macula-off group at 6 months after SB surgery. DCP has been shown to be more vulnerable to hypoxic injury [14]. Vascular alterations of this layer are typical of chronic diseases, such as diabetic retinopathy [24,25]. In the context of retinal detachment, a detached macula usually means a less recent onset compared with macula-on RRDs. The fact that DCP VD is reduced in the macula-off group could be secondary either to damage induced by the subretinal fluid or to microvascular alterations related to a longer disease duration. The increase in DCP VD during follow-up might indicate that successful surgery removed the causative factor of such microvascular injury. However, the understanding of this mechanism is limited by the lack of baseline scans.

Several limitations characterize the present study. First, the retrospective design might have introduced some bias. However, strict eligibility criteria should have reduced this risk. Second, the sample size was relatively small. A larger population would have allowed additional analyses and further investigation of which clinical variables might have an influence on VD. Nonetheless, we included a total of 37 eyes and all surgeries were performed by the same vitreoretinal consultant. With regard to OCTA imaging, our software did not provide two different values of FAZ area for the deep and superficial capillary plexi, but only one value, including both layers. Furthermore, no correlation analysis was conducted in the macula-off group. The reason for this is that in this group, no baseline OCTA images were available because of the macula-off status. A correlation analysis between VD change and other variables would have been of poor reliability, given the absence of baseline OCTA data.

## 5. Conclusions

Our analyses showed that VD on OCTA was reduced following SB surgery for RRD. These microvascular alterations, to some extent, are similar to those reported after vitrectomy and could be a consequence of the disease rather than the surgical procedure. Axial length increase seems to be associated with worse VD outcome. Whether this is secondary to the presence of a greater amount of subretinal fluid at baseline or to mechanical damage caused by the encircling band needs to be further investigated. Larger studies are warranted to better understand which mechanisms lead to such microvascular impairment and which clinical variables might have an influence on it.

## Figures and Tables

**Figure 1 diagnostics-12-03015-f001:**
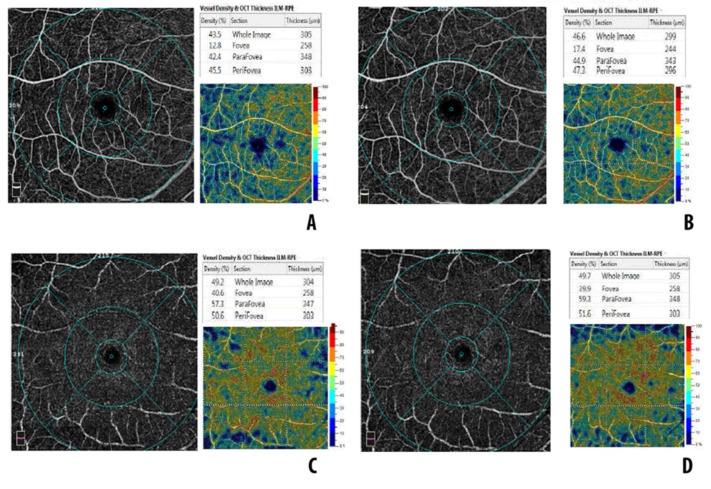
OCTA imaging in a case of macula-on retinal detachment: (**A**) baseline superficial capillary plexus; (**B**) superficial capillary plexus at 6 months postoperatively; (**C**) baseline deep capillary plexus; (**D**) deep capillary plexus at 6 months postoperatively.

**Figure 2 diagnostics-12-03015-f002:**
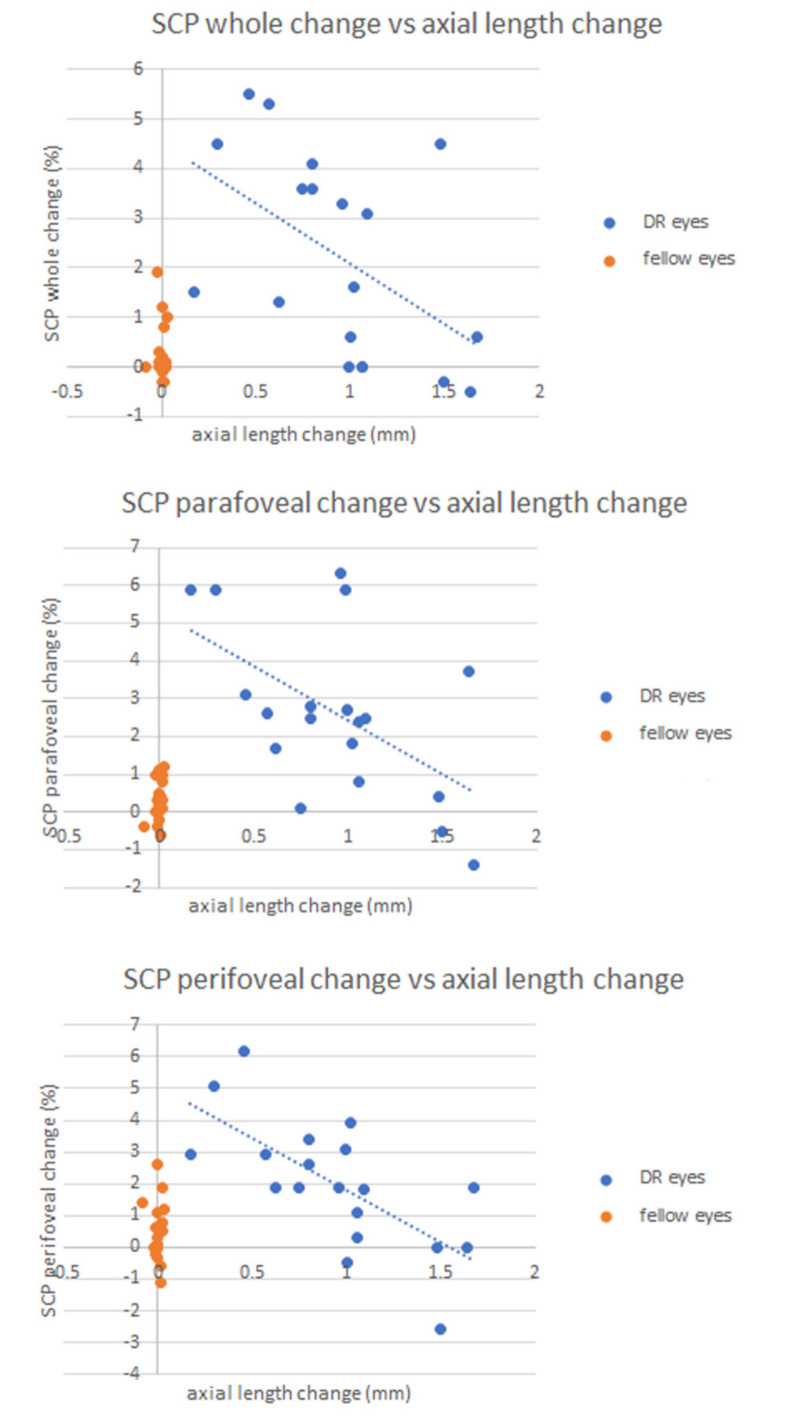
Scatterplots illustrating correlation analysis in the macula-on group between axial length change and vessel density change in the whole parafoveal and perifoveal superficial capillary plexi (SCP). Blue dots represent retinal detachment eyes (DR eyes), while orange dots represent fellow eyes.

**Figure 3 diagnostics-12-03015-f003:**
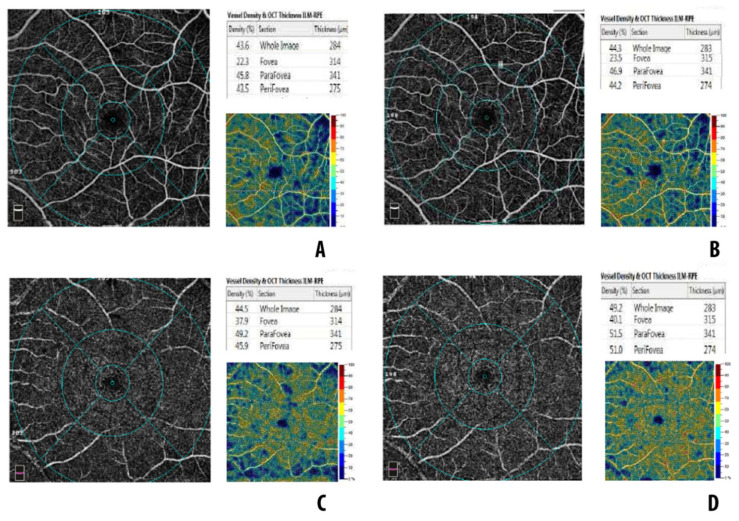
OCTA imaging in a case of macula-off retinal detachment: (**A**) superficial capillary plexus at 1 month postoperatively; (**B**) superficial capillary plexus at 6 months postoperatively; (**C**) deep capillary plexus at 1 month postoperatively; (**D**) deep capillary plexus at 6 months postoperatively.

**Table 1 diagnostics-12-03015-t001:** Baseline demographic and ocular characteristics of the included patients.

	Macula-On Group	Macula-Off Group	*p*
**Eyes (*n*.)**	19	18	
**Gender (male/female)**	9/10	10/8	0.862
**Age, years**	54 ± 6 (45–63)	56 ± 7 (43–65)	0.356
**BCVA, logMAR**	0.10 ± 0.03 (0.04–0.16)	0.94 ± 0.07 (0.88–1.20)	<0.001
**Axial length, mm**	25.2 ± 0.5 (24.23–25.92)	24.8 ± 1.1 (23.76–25.81)	0.101

Footnote: Data are expressed as mean ± SD (range of minimum and maximum values); BCVA: best-corrected visual acuity; logMAR: logarithm of the minimum angle of resolution; mm = millimeters.

**Table 2 diagnostics-12-03015-t002:** Clinical and OCTA parameters of the macula-on RRD group.

	Macula-On RRD Eyes, *n* 19					*Fellow Eyes, n 19*				*t Test*			
	Baseline	1 Month	3 Months	6 Months	*p (ANOVA)*	Baseline	1 Month	3 Months	6 Months	Baseline	1 Month	3 Months	6 Months
**BCVA (logMar)**	0.10 ± 0.03 (0.04–0.16)	0.09 ± 0.04 (0.02–0.14)	0.09 ± 0.05 (0.02–0.14)	0.09 ± 0.03 (0.04–0.16)	*0.687*	0.07 ± 0.03 (0.02–0.10)	0.07 ± 0.04 (0.00–0.14)	0.08 ± 0.03 (0.02–0.14)	0.08 ± 0.04 (0.02–0.16)	*0.003*	*0.081*	*0.187*	*0.062*
**axial length (mm)**	25.2 ± 0.5 (24.23–25.92)	-	-	26.1 ± 0.8 (24.77–27.32)	*<0.001 (t-test)*	26.0 ± 1.5 (24.23–27.83)	-	-	26.0 ± 1.3 (24.25–27.83)	*0.014*	*-*	*-*	*0.475*
**FAZ**	0.227 ± 0.019 (0.181–0.261)	0.230 ±0.007 (0.217–0.242)	0.221 ± 0.015 0.181–0.241)	0.229 ± 0.016 (0.193–0.261)	*0.310*	0.226 ± 0.014 (0.194–0.254)	0.226 ± 0.011 (0.205–0.244)	0.221 ± 0.013 (0.189–0.233)	0.224 ± 0.014 (0.181–0.244)	*0.863*	*0.084*	*0.983*	*0.164*
**CMT (µm)**	244 ± 13 (229–270)	250 ± 10 (236–272)	252 ± 9 (231–271)	249 ± 11 (229–273)	*0.197*	246 ± 12 (229–266)	249 ± 13 (231–273)	248 ± 12 (238–274)	247 ± 12 (231–273)	*0.612*	*0.203*	*0.091*	*0.183*
**Vessel density**													
**SCP whole**	40.9 ± 1.8 ^c^ (37.8–43.3)	42.1 ± 2.2 (38.6–45.3)	42.6 ± 1.6 ^c^ (38.6–45.3)	43.1 ± 2.2 ^d^ (39.4–46.7)	*0.004*	44.3 ± 1.6 (40.2–46.1)	44.4 ± 1.8 (40.3–48.1)	44.2 ± 1.5 (40.7–46.3)	44.2 ± 1.9 (39.9–46.4)	*<0.001*	*<0.001*	*<0.001*	*0.003*
**SCP fovea**	21.4 ± 2.8 (12.8–27.8)	20.1 ± 2.9 (9–22.6)	19.8 ± 1.4 (17.2–22.2)	21.3 ± 1.7 (17.4–23.8)	*0.073*	21.5 ± 1.7 (17.2–23.8)	21.3 ± 1.2 (19.3–23.7)	21.4 ± 1.3 (18.9–23.2)	21.6 ± 1.6 (17.6–23.7)	*0.826*	*0.027*	*0.012*	*0.121*
**SCP parafovea**	41.0 ± 2.6 (37.5–46.5)	43.1 ± 1.9 (39.7–46.5)	43.5 ± 1.7 ^e^ (39.2–46.3)	43.6 ± 2.8 ^f^ (36.8–49.4)	*0.005*	45.6 ± 1.5 (43.1–49.1)	45.8 ± 1.7 (43.7–50.8)	45.5 ± 1.4 (43.5–48.5)	45.2 ± 1.4 (42.1–48.7)	*<0.001*	*<0.001*	*<0.001*	*0.006*
**SCP perifovea**	40.7 ± 2.4 (35.9–45.5)	39.9 ± 1.4 (37.2–43.6)	40.6 ± 1.4 (37.6–42.8)	42.7 ± 2.8 ^g^ (37.9–48.8)	*0.001*	44.4 ± 1.8 (41.7–47.2)	44.2 ± 2.1 (40.1–48.1)	44.3 ± 1.7 (41.2–47.1)	44.1 ± 1.6 (41.1–46.9)	*<0.001*	*<0.001*	*<0.001*	*0.021*
**DCP whole**	48.8 ± 1.8 (42.9–51.1)	49.7 ± 1.3 (46.9–51.9)	49.4 ± 1.7 (46.1–52.9)	49.4 ± 2.1 (43.9–53.3)	*0.481*	50.1 ± 2.1 (44.8–53.4)	50.1 ± 1.9 (47.5–54.2)	49.7 ± 2.1 (44.9–52.6)	49.9 ± 1.9 (45.1–52.2)	*0.065*	*0.105*	*0.184*	*0.109*
**DCP fovea**	40.7 ± 1.4 (38.2–44.2)	40.2 ± 1.7 (37.5–44.3)	40.9 ± 2.0 (38.4–44.3)	41.1 ± 1.2 (38.1–43.3)	*0.320*	41.6 ± 1.5 (38.0–44.1)	41.5 ± 1.2 (39.1–43.7)	41.5 ± 1.3 (38.5–43.7)	41.6 ± 1.8 (37.8–46.5)	*0.078*	*0.055*	*0.091*	*0.207*
**DCP parafovea**	52.1 ± 2.3 (47.3–57.3)	51.1 ± 1.6 (47.3–53.6)	50.8 ± 2.2 (46.0–55.8)	51.4 ± 2.7 (46.4–59.3)	*0.320*	52.6 ± 2.1 (47.9–57.7)	52.4 ± 1.9 (48.2–56.7)	52.1 ± 23.3 (48.1–55.9)	52.3 ± 2.1 (43.3–54.4)	*0.530*	*0.093*	*0.059*	*0.087*
**DCP perifovea**	48.2 ± 2.9 (42.2–53.4)	49.2 ± 1.8 (46.6–53.4)	47.8 ± 2.8 (41.9–53.4)	48.4 ± 3.1 (42.0–53.4)	*0.432*	49.4 ± 2.9 (43.3–54.7)	49.6 ± 2.6 (45.3–54.9)	49.1 ± 2.7 (42.9–53.2)	49.4 ± 2.7 (43.5–53.7)	*0.200*	*0.481*	*0.183*	*0.367*

Footnote: Data are expressed as mean ± SD (range of minimum and maximum values); OCTA, optical coherence tomography angiography; RRD, rhegmatogenous retinal detachment; *n*, number; BCVA, best corrected visual acuity; logMar, logarithm of minimum angle of resolution; FAZ, foveal avascular zone; CMT. Central Macular Thickness; SCP, superficial capillary plexus; DCP, deep capillary plexus; *p* (Tukey HSD vs. baseline) c = 0.040, d = 0.003, e = 0.012, f = 0.008, g = 0.021.

**Table 3 diagnostics-12-03015-t003:** Clinical and OCTA parameters of the macula-off RRD group.

	Macula-Off RRD Eyes, *n* 19					*Fellow Eyes, n 19*				*t Test*			
	Baseline	1 Month	3 Months	6 Months	*p (ANOVA)*	Baseline	1 Month	3 Months	6 Months	Baseline	1 Month	3 Months	6 Months
**BCVA (logMar)**	1.04 ± 0.09 (0.88–1.20)	0.30 ± 0.10 (0.12–0.46)	0.26 ± 0.08 (0.12–0.38)	0.25 ± 0.08 (0.12–0.38)	*<0.001*	0.09 ± 0.04 (0.2–0.12)	0.09 ± 0.05 (0.2–0.20)	0.08 ± 0.04 (0.2–0.18)	0.08 ± 0.04 (0.4–0.20)	*<0.001*	*<0.001*	*<0.001*	*<0.001*
**axial length (mm)**	24.9 ± 0.5 (24.31–25.8)			26.2 ± 0.7 (24.77–27.32)	*0.001 (t-test)*	25.6 ± 0.8 (24.23–27.83)			25.6 ± 0.8 (24.25–27.81)	*0.019*			*0.037*
**FAZ (mm2)**	NA	0.231 ± 0.031 (0.166–0.288)	0.218 ± 0.017 (0.171–0.242)	0.226 ± 0.013 (0.173–0.263)	*0.224*	0.231 ± 0.010 (0.216–0.254)	0.230 ± 0.016 (0.193–0.267)	0.228 ± 0.012 (0.195–0.249)	0.232 ± 0.015 (0.219–0.281)	*-*	*0.744*	*0.061*	*0.186*
**CMT (µm)**	NA	273 ± 14 (259–314)	276 ± 13 (249–296)	275 ± 15 (249–315)	*0.316*	269 ± 11 (249–294)	267 ± 8 (252–285)	271 ± 9 (259–289)	268 ± 11 (248–288)	*-*	*0.289*	*0.227*	*0.194*
**Vessel density**													
**SCP whole**	NA	40.5 ± 1.4 (38.6–43.6)	40.6 ± 1.3 (38.6–42.6)	41.3 ± 1.7 (36.9–44.3)	*0.054*	44.6 ± 1.1 (42.6–47.3)	44.7 ± 1.3 (42.5–46.8)	44.7 ± 1.5 (42.6–48.3)	44.7 ± 1.5 (42.4–48.3)	*-*	*<0.001*	*<0.001*	*<0.001*
**SCP fovea**	NA	20.1 ± 1.4 (17.2–22.3)	20.5 ± 1.0 (18.3–22.3)	20.9 ± 1.6 (17.2–23.7)	*0.108*	20.5 ± 1.6 (17.2–23.8)	20.6 ± 1.6 (17.5–24.1)	20.4 ± 1.5 (17.2–23.8)	20.7 ± 1.5 (18.1–24.2)	*-*	*0.532*	*0.871*	*0.718*
**SCP parafovea**	NA	41.5 ± 1.6 (39.2–45.8)	41.6 ± 1.3 (39.2–44.8)	41.6 ± 1.9 (38.1–46.9)	*0.689*	45.1 ± 1.5 (42.7–47.6)	45.3 ± 1.5 (42.9–47.9)	45.2 ± 1.4 (42.5–47.8)	45.2 ± 1.6 (42.3–48.1)	*-*	*<0.001*	*<0.001*	*<0.001*
**SCP perifovea**	NA	40.3 ± 1.7 (37.2–43.6)	40.6 ± 1.4 (37.6–42.8)	41.0 ± 1.5 (37.6–46.9)	*0.108*	43.5 ± 1.3 (41.4–46.2)	43.6 ± 1.4 (41.7–46.5)	43.4 ± 1.3 (41.1–47.3)	43.6 ± 1.5 (40.9–46.4)	*-*	*<0.001*	*<0.001*	*<0.001*
**DCP whole**	NA	41.0 ± 1.6 (38.6–44.5)	43.6 ± 1.2 (40.8–45.3)	43.9 ± 1.7 (41.8–49.2)	*<0.001*	46.5 ± 1.2 (44.2–48.2)	46.6 ± 1.1 (45.1–48.6)	46.4 ± 1.2 (44.6–48.7)	46.5 ± 1.2 (43.9–48.4)	*-*	*<0.001*	*<0.001*	*<0.001*
**DCP fovea**	NA	35.4 ± 2.4 (32.6–44.5)	36.3 ± 1.9 (33.2–41.3)	36.2 ± 1.8 (33.8–40.1)	*0.068*	37.7 ± 1.0 (36.1–39.4)	37.8 ± 1.0 (36.5–39.7)	37.5 ± 1.1 (36.4–39.1)	37.6 ± 1.0 (35.3–39.2)	*-*	*<0.001*	*0.016*	*0.009*
**DCP parafovea**	NA	45.0 ± 1.4 (42.9–49.2)	46.1 ± 1.7 (41.8–48.3)	48.6 ± 2.6 (42.1–52.9)	*<0.001*	51.8 ± 2.1 (47.9–57.7)	51.9 ± 2.0 (49.2–57.9)	51.6 ± 2.1 (48.5–58.2)	51.5 ± 2.4 (45.8–56.9)	*-*	*<0.001*	*<0.001*	*<0.001*
**DCP perifovea**	NA	40.4 ± 2.1 (36.5–45.9)	40.9 ± 1.5 (37.6–43.1)	40.5 ± 2.9 (37.6–51.1)	*0.317*	47.2 ± 1.6 (44.5–51.6)	47.2 ± 1.7 (44.3–51.7)	47.4 ± 1.8 (44.4–52.9)	47.3 ± 1.5 (44.8–51.4)	*-*	*<0.001*	*<0.001*	*<0.001*

Footnote: Data are expressed as mean ± SD (range of minimum and maximum values); OCTA, optical coherence tomography angiography; RRD, rhegmatogenous retinal detachment; *n*, number; BCVA, best corrected visual acuity; logMar, logarithm of minimum angle of resolution; FAZ, foveal avascular zone; CMT, central macular thickness; SCP, superficial capillary plexus; DCP, deep capillary plexus; NA, not applicable.

## Data Availability

Not applicable.

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
