# Peer review of "Microvascular Changes after Scleral Buckling for Rhegmatogenous Retinal Detachment: An Optical Coherence Tomography Angiography Study"

_diagnostics, 2022, doi:10.3390/diagnostics12123015_

Round 1

Reviewer 1 Report

In the study carried out, “Microvascular Changes After Scleral Buckling For Rhegmatogenous Retinal Detachment: An Optical Coherence Tomography Angiography Study” it is interesting and necessary for the correct interpretation of the vessel density of both plexuses (SCP and DPC) measured with OCTA after a rhegmatogenous retinal detachment treated with scleral buckling, both with macula-on and macula-off, using the fellow eye of each patient as a control.

The aims of the study are satisfied during the results, discussion and conclusion, ending that “Our analyses showed that vessel density on OCTA is reduced following scleral buck- ling surgery for RRD. These microvascular alterations, to some extent, are similar to those reported after vitrectomy and could be a consequence of the disease rather than the surgical procedure. Axial length increase seems to be associated with worse vessel density outcome. Whether this is secondary to the presence of a greater amount of subretinal fluid at baseline or to mechanical damage caused by the encircling band needs to be further investigated.” They said that more studies are needed, I agree but I consider that in those new future studies more participants, in both the macula-on and macula-off groups, are also needed to consolidate the results obtained in this study.

To improve the article, I propose the following details to the authors:

The abstract is well written and summarizes what has been studied quite well. In the keywords I suggest adding vessel density, superficial capillary plexus and deep capillary plexus.

Introduction:

Vessel density (VD) appears abbreviated in the abstract but here it appears without abbreviating, unifying, since this happens again in the discussion. I suggest writing continuously abbreviated (VD). (Last paragraph).

Materials and methods:

Add the approval number of the research ethics committee.

They comment that: “These exclusion criteria were applied to the fellow eye as well and patients with an anisometropia > 2.0 Diopters were excluded since the fellow eye was used as the control eye.”

I see that the axial length less than 26mm was taken into account. If the axial length is very high, there are also changes in the retinal microvasculature. Considering that retinal detachments are more common in myopic eyes with long axial lengths, which refractive limit in terms of spherical equivalent was considered as inclusion criteria in the study?

First line of page 3 vessel density unabbreviated and in the next paragraph abreviated.

Define FAZ in the paragraph before “Statistical analysis” for the first use.

Results:

The results are well presented, but it is helpful to put the ranges with the maximum value and the minimum value in each parameter of the tables to see in which ranges the sample falls.

A figure 4 like figure 2 would be useful for the macula-off group for months 1-6.

Discussion and conclusions:

The discussion is consistent and compares the results obtained with other studies, taking into account the limitations that have occurred in the present study.

Vessel density (VD) appears continuously without abbreviation, but occasionally abbreviated. Unify using the abbreviated term to facilitate reading.

The same thing happens with SCP and DCP and with SB. Unify to abbreviated.

When citing an article add a point in "et al."

When talking about the study of OCTA in diabetic retinopathy, I propose to cite another article next to reference 24 where a study of the vessel density in different plexuses has been carried out, published in this same journal.

Fernández-Espinosa G, Boned-Murillo A, Orduna-Hospital E, Díaz-Barreda MD, Sánchez-Cano A, Bielsa-Alonso S, Acha J, Pinilla I. Retinal Vascularization Abnormalities Studied by Optical Coherence Tomography Angiography (OCTA) in Type 2 Diabetic Patients with Moderate Diabetic Retinopathy. Diagnostics (Basel). 2022 Feb 1;12(2):379. doi: 10.3390/diagnostics12020379. PMID: 35204470; PMCID: PMC8871460.

Author Response

Reviewer 1

In the study carried out, “Microvascular Changes After Scleral Buckling For Rhegmatogenous Retinal Detachment: An Optical Coherence Tomography Angiography Study” it is interesting and necessary for the correct interpretation of the vessel density of both plexuses (SCP and DPC) measured with OCTA after a rhegmatogenous retinal detachment treated with scleral buckling, both with macula-on and macula-off, using the fellow eye of each patient as a control.

The aims of the study are satisfied during the results, discussion and conclusion, ending that “Our analyses showed that vessel density on OCTA is reduced following scleral buck- ling surgery for RRD. These microvascular alterations, to some extent, are similar to those reported after vitrectomy and could be a consequence of the disease rather than the surgical procedure. Axial length increase seems to be associated with worse vessel density outcome. Whether this is secondary to the presence of a greater amount of subretinal fluid at baseline or to mechanical damage caused by the encircling band needs to be further investigated.” They said that more studies are needed, I agree but I consider that in those new future studies more participants, in both the macula-on and macula-off groups, are also needed to consolidate the results obtained in this study.

To improve the article, I propose the following details to the authors:

The abstract is well written and summarizes what has been studied quite well. In the keywords I suggest adding vessel density, superficial capillary plexus and deep capillary plexus.

Response

We are grateful to Reviewer 1 for his/her comments. Following your suggestion, these keywords have been added in the revised version of the manuscript.

Introduction:

Vessel density (VD) appears abbreviated in the abstract but here it appears without abbreviating, unifying, since this happens again in the discussion. I suggest writing continuously abbreviated (VD). (Last paragraph).

Response

Thank you for your comment. We have now used VD wherever possible.

Materials and methods:

Add the approval number of the research ethics committee.

Response

Thank you for your comment. This has been done. Page 2, line 57.

They comment that: “These exclusion criteria were applied to the fellow eye as well and patients with an anisometropia > 2.0 Diopters were excluded since the fellow eye was used as the control eye.”

I see that the axial length less than 26mm was taken into account. If the axial length is very high, there are also changes in the retinal microvasculature. Considering that retinal detachments are more common in myopic eyes with long axial lengths, which refractive limit in terms of spherical equivalent was considered as inclusion criteria in the study?

Response

Thank you for your comment. A spherical equivalent ≥ -6 Diopters was considered as an exclusion criterion. Following your suggestion, this has been added in the revised version of the manuscript (Methods, page 2, lines 72-73).

First line of page 3 vessel density unabbreviated and in the next paragraph abreviated.

Response

Following your suggestion, the acronym VD has been used throughout the manuscript.

Define FAZ in the paragraph before “Statistical analysis” for the first use.

Response

Thank you for your comment. The acronym FAZ was put in brackets where foveal avascular zone was spelled out for the very first time. Page 3, lines 101-102

Results:

The results are well presented, but it is helpful to put the ranges with the maximum value and the minimum value in each parameter of the tables to see in which ranges the sample falls.

Response

Thank you for your suggestion. Ranges with minimum and maximum values have been added in the revised version of both Tables 2 and 3.

A figure 4 like figure 2 would be useful for the macula-off group for months 1-6.

Response

Thank you for your comment. Unfortunately, no correlation analysis was conducted in the macula-off group. The reason is the fact that in the macula-off group there were no baseline OCTA images because of the macula-off status. Thus, a correlation analysis between VD change and other variables would have been of poor reliability given the absence of baseline OCTA data. This has been now better clarified in the main text among study limitations. Page 14, lines 339-343. (“Furthermore, no correlation analysis was conducted in the macula-off group. The reason is that in this group no baseline OCTA images were available because of the macula-off status. A correlation analysis between VD change and other variables would have been of poor reliability given the absence of baseline OCTA data”)

Discussion and conclusions:

The discussion is consistent and compares the results obtained with other studies, taking into account the limitations that have occurred in the present study.

Vessel density (VD) appears continuously without abbreviation, but occasionally abbreviated. Unify using the abbreviated term to facilitate reading.

 Response

Thank you. This has been done.

The same thing happens with SCP and DCP and with SB. Unify to abbreviated.

Response

Thank you. This has been done

When citing an article add a point in "et al."

Response

Thank you. This has been done

When talking about the study of OCTA in diabetic retinopathy, I propose to cite another article next to reference 24 where a study of the vessel density in different plexuses has been carried out, published in this same journal.

Fernández-Espinosa G, Boned-Murillo A, Orduna-Hospital E, Díaz-Barreda MD, Sánchez-Cano A, Bielsa-Alonso S, Acha J, Pinilla I. Retinal Vascularization Abnormalities Studied by Optical Coherence Tomography Angiography (OCTA) in Type 2 Diabetic Patients with Moderate Diabetic Retinopathy. Diagnostics (Basel). 2022 Feb 1;12(2):379. doi: 10.3390/diagnostics12020379. PMID: 35204470; PMCID: PMC8871460.

Response

Thank you for your comment. This reference has been added as number 25.  

Reviewer 2 Report

In this manuscript by Fallico et al., the authors describe a retrospective study to assess changes in macular vasculature post-surgery for retinal detachment. This study focused on patients that underwent scleral buckling and collected data for both macula-on and macula-off groups. Optical coherence tomography angiography was used for microvasculature assessment at the time of surgery and 6 months post-surgery. Overall this is a well-written manuscript, however, the significance of this work is not very clear. The comments are as follows:

1.       The authors do not clearly define what they consider as baseline, especially for eye with retinal detachment.

2.       Was the control eye also followed for measurements at 6-months? In the current manuscript the information suggests only baseline measurement was done for the control eye. It is ideal to collect control eye measurements as well when collecting measurements for affected eye.

3.       Figure 2: Please include a separate graph or use different colored dots on the current graphs to show data compared to control. Based on the numbers listed in the tables, it appears that most of the measurements in the affected eyes are trending towards control baseline measurements.

4.       It would be good to include scatterplots for macula-off group similar to as shown for macula-on group in Figure 2.

5.       In third graph for Figure 2, length is misspelled for X-axis.

Author Response

Reviewer 2

In this manuscript by Fallico et al., the authors describe a retrospective study to assess changes in macular vasculature post-surgery for retinal detachment. This study focused on patients that underwent scleral buckling and collected data for both macula-on and macula-off groups. Optical coherence tomography angiography was used for microvasculature assessment at the time of surgery and 6 months post-surgery. Overall this is a well-written manuscript, however, the significance of this work is not very clear. The comments are as follows:

  1. The authors do not clearly define what they consider as baseline, especially for eye with retinal detachment.

Response

We are grateful to Reviewer#2 for his/her comment. Baseline was considered as the time-point before scleral buckling surgery. This has been better clarified in methods section. All patients received a complete eye examination at baseline, which means before SB surgery, and at each follow-up visit, namely at 1, 3 and 6 months postoperatively. (Page 2, lines 83-84)

  1. Was the control eye also followed for measurements at 6-months? In the current manuscript the information suggests only baseline measurement was done for the control eye. It is ideal to collect control eye measurements as well when collecting measurements for affected eye.

Response

Thank you for your comment. The fellow eye was followed up for 6 months as well. We have now added in the revised version of both Table 2 and Table 3 data on all follow-ups of fellow eyes.

  1. Figure 2: Please include a separate graph or use different colored dots on the current graphs to show data compared to control. Based on the numbers listed in the tables, it appears that most of the measurements in the affected eyes are trending towards control baseline measurements.

Response

Thank you for your comment.  Following your suggestion, we have included in the same graph data on control group, which are represented by orange dots.

  1. It would be good to include scatterplots for macula-off group similar to as shown for macula-on group in Figure 2.

Response

Thank you for your comment. Figure 2 displays scatterplots illustrating correlation analyses (between VD change and other variables) in the macula-on group. Unfortunately, no correlation analysis was conducted in the macula-off group. The reason is the fact that in the macula-off group there were no baseline OCTA images because of the macula-off status. Thus, a correlation analysis between VD change and other variables would have been of poor reliability given the absence of baseline OCTA data. This has been now better clarified in the main text among study limitations. Page 14, lines 339-343. (“Furthermore, no correlation analysis was conducted in the macula-off group. The reason is that in this group no baseline OCTA images were available because of the macula-off status. A correlation analysis between VD change and other variables would have been of poor reliability given the absence of baseline OCTA data”)

  1. In third graph for Figure 2, length is misspelled for X-axis.

Response

This has been done.

Round 2

Reviewer 1 Report

Dear authors,

It is a novelty well-written study where it meets the objectives proposed throughout it. The methods are well described and, based on the results they obtain, they reach well-supported conclusions.

I consider that the answers and corrections are done and that the article is now complete.

I point out two things:

-   In the article the sentence is incomplete, although it is complete in the answers:

(axial length > 26 mm and/or a spherical equiva-72 lent XXXX)

- I suggest to put the ranges with the maximum value and the minimum value in each parameter of the table 1.

Author Response

Dear Reviewer#1, 

Thank you for your comments. 

1) In the article the sentence is incomplete, although it is complete in the answers:

(axial length > 26 mm and/or a spherical equivalent ≥XXXX)

Response: 

This sentence has been completed in the manuscript. Thank you for spotting this mistake. 

2) I suggest to put the ranges with the maximum value and the minimum value in each parameter of the table 1.

Response: 

Thank you for your comment. Table 1 has now been revised and ranges with minimum and maximum values have been added.